# Halochromic Textiles for Real-Time Sensing of Hazardous Chemicals and Personal Protection

**DOI:** 10.3390/ma16082938

**Published:** 2023-04-07

**Authors:** Liliana Leite, Vânia Pais, Cristina Silva, Inês Boticas, João Bessa, Fernando Cunha, Cátia Relvas, Noel Ferreira, Raul Fangueiro

**Affiliations:** 1Fibrenamics—Institute of Innovation on Fiber-Based Materials and Composites, University of Minho, 4800-058 Guimarães, Portugal; 2Centre for Textile Science and Technology (2C2T), University of Minho, 4800-058 Guimarães, Portugal; 3A. Ferreira & Filhos, Rua Amaro de Sousa 408, 4815-901 Caldas de Vizela, Portugal; 4Department of Textile Engineering, University of Minho, 4800-058 Guimarães, Portugal

**Keywords:** defense materials, personal protection, chemical protective clothing (CPC), halochromic textiles, hazardous chemicals, pH indicators

## Abstract

Chemical protective clothing (CPC) has become mandatory when performing various tasks to ensure user protection and prevent chemicals from contacting the skin and causing severe injuries. In addition to protection, there is a need to develop a simple mechanism that can be attached to CPC and be capable of detecting and alerting the user to the presence of harmful chemical agents. In this study, a double-sensor strategy was investigated, using six different pH indicators stamped on cotton and polyester knits to detect acidic and alkaline substances, both liquid and gaseous. Functionalized knits underwent microscopic characterization, air permeability and contact angle evaluation. All samples exhibited hydrophobic behavior (contact angle > 90°) and air permeability values above 2400 L/min/cm^2^/bar, with the best condition demonstrating a contact angle of 123° and an air permeability of 2412.5 L/min/cm^2^/bar when the sensor methyl orange and bromocresol purple (MO:BP) was stamped on polyester. The performed tests proved the functionality of the sensors and showed a visible response of all knits when contacting with different chemicals (acids and bases). Polyester functionalized with MO:BP showed the greatest potential, due to its preeminent color change. Herein, the fiber coating process was optimized, enabling the industrial application of the sensors via a stamping method, an alternative to other time- and resource-consuming techniques.

## 1. Introduction

In recent years, it has become mandatory to use protective equipment when performing several functions. From this need, several studies have emerged around technical textiles and fibers capable of responding to the demands of non-conventional textile markets [1], such as personal protection. Chemical protective clothing (CPC) is equipment made to be the last line of defense against hazardous compounds, preventing their contact with the skin and severe injuries [2]. CPC is essential in many areas where there is a risk of exposure, such as military and law enforcement personnel, laboratory workers or chemical industries [3]. For the last two scenarios, chemical protective clothing must be used whenever there is a possible danger, which can be during production, distribution, storage and use. These chemicals may be gases, liquids, or solids [4]. Moreover, military soldiers or law enforcement officers use chemical protection for initial entry, victim rescue, evidence collection, body removal and in biochemical/terrorism attacks. During the development of such clothing, it is important to obtain a protection barrier without compromising comfort, i.e., it is desirable that the equipment is not only hydrophobic, to prevent chemicals from being absorbed into the textile, but also permeable to air and moisture to guarantee user well-being and thermo-physiological comfort [5].

In addition to protection capability, another inherent need for personal protective equipment (PPE) is related to the ability of the equipment itself to detect and alert the user to the presence of a certain chemical agent that may pose a risk to his health [6]. Furthermore, detection systems for this type of substance are mainly electronic, such as the AP4C models, HazMat ID, CAM and SAW MINICAD [7]. However, the complexity and volume of these devices do not always make them suitable for field application because they either require the use of specialized equipment and well-trained technicians or cannot be directly integrated into protective clothing and worn by professionals during their most routine activities.

Other approaches that can be used for chemical sensing are based on a change in the optical response, for example, using fluorometric probes [8,9,10] and halochromic molecules [11,12,13,14]. Halochromic molecules are those capable of shifting colors in response to a variation in pH [15]. The literature describes several dyes that have this responsive behavior, for example, methyl orange, methyl red, bromocresol purple, bromocresol green, bromothymol blue, brilliant yellow and phenol red [16,17]. Even though the systems based on pH-sensitive compounds lack specificity, they are cheap, easy to produce and generate a rapid color change, which allows the operator to react quickly in case of exposure and does not require advanced training for equipment manipulation. This type of technology has been applied in textiles already by some research groups. For example, Stojkoski and Kert developed a pH-sensitive polyamide fabric using Bromocresol green, which had the purpose of detecting the pH value of rainwater and the presence of air pollutants [18]. Van der Schueren and De Clerck proved the viability of dyeing cotton and nylon with pH indicators by analyzing different compounds, particularly Brilliant Yellow and Alizarin, and their color depth, levelness, exhaustion and color response to different pH on the textiles [19]. Furthermore, Park et al. applied a screen-printing method to produce a polyester textile sensitive to acidic (HCl) and basic (NH_3_) atmospheres, even at low concentrations [20]. The chemical detection was done using two halochromic sensors, Dye 3 and RhYK.

As seen, there is a deficit of PPE that incorporates a simple, immediate and instinctive alert system, when faced with an external chemical stimulus. This work aims to develop cotton and polyester textiles capable of sensing acidic and alkaline substances, both liquid and gaseous. These products can then be attached to CPC in the form of patches in strategically selected places such as on the sleeve, near the cuff and on the chest. This selection was made based on the most likely points of contact with harmful agents, by spills or splashes, as well as the user’s ease of observation, allowing for quick detection of the threat and subsequent response. Cotton and polyester were chosen because they are the most commonly used natural and synthetic fibers [21,22,23], allowing the effectiveness of the functional formulation to be tested on different substrates.

Hence, the authors intend to study the optimization of the process of fiber coating, by using a stamping method, which is quicker than dyeing, one of the most conventional methods used for this application [24,25,26], positively impacting production time and resource consumption (i.e., water). The optimal pH during the sensor solution production was determined and the samples obtained were characterized to ensure they displayed proper functional and structural behavior for personal protection applications. For the detection a double sensor approach was tested, using two halochromic compounds with sensitivity for different areas of the pH scale. Based on previous research in other areas of study, this method allows for higher accuracy of results while avoiding false positives [27,28]. Bromothymol blue (BB), methyl red (MR), methyl red sodium (MR), methyl orange (MO) and bromocresol purple (BP) were the pH indicators used in this study.

## 2. Materials and Methods

### 2.1. Materials

The cotton and polyester jersey knits were provided by A. Ferreira & Filhos. One knit was composed of 100% polyester with a thickness of 0.68 mm and a mass per unit area of 207 g/m^2^ and the other of 100% cotton with a thickness of 0.89 mm and a mass per unit area of 187 g/m^2^. The pH indicators used for the development of the sensors were methyl red (C_15_H_15_N_3_O_2_), methyl red sodium salt (C_15_H_15_N_3_O_2_Na), methyl orange (C_14_H_14_N_3_NaO_3_S), bromocresol purple (C_21_H_16_Br_2_O_5_S), bromothymol blue (C_27_H_28_Br_2_O_5_S) and were purchased from Sigma-Aldrich (St. Louis, Mo., EUA). The chemicals used in the evaluation of samples sensitivity were the following: acetic acid acquired from Chem-lab (Zedelgem, Belgium); sodium hydroxide purchased from Normax (Marinha Grande, Portugal); formic acid obtained from Fisher Scientific (Hampton, NH., EUA); ammonia purchased from VWR Chemicals (Radnor, PA, EUA); and citric acid acquired from Scharlau (Barcelona, Spain). The polyurethane polymeric base employed in the preparation of pH-sensitive formulations before application on fibrous substrates, Edolan SN and Thickener A02 were purchased from ADI Center Portugal, S.A (Santo Tirso, Portugal). Glycerol was obtained from Scharlau. The filter paper, used as substrate in one part of this work, was purchased from Normax.

### 2.2. Preparation of pH-Sensitive Aqueous Solutions and Application on Paper Substrates

Based on previous research [22], the indicators bromothymol blue (BB), methyl red (MR), methyl red sodium (MR), methyl orange (MO) and bromocresol purple (BP) were chosen. To assess the detection performance of these sensors and determine the optimal pH during the production of the sensor solution, an initial study was conducted using filter paper samples functionalized with aqueous formulations doped with pH indicators. The three sensors employed were: methyl red (0.48 g/L) and methyl red sodium salt (0.51 g/L) (MR:MR); methyl red (0.6 g/L) and bromothymol blue (1 g/L) (MR:BB); methyl orange (0.51 g/L) and bromocresol purple (1 g/L) (MO:BP). The solutions were prepared with distilled water and glycerol 16% (*v*/*v*). After mixing these different reagents with a magnetic stirrer, the solutions were placed in Petri dishes with the substrate (filter paper). After 1 h of being submerged, the filter paper was removed and dried at room temperature. To verify which was the ideal original pH for the sensors and to evaluate if this parameter would affect the response time and sensing capacity, the three previously selected dual sensors were prepared at pH 4, 7 and 11.

### 2.3. Evaluation of Paper Substrates’ Sensitivity to Liquid Chemicals

Taking into consideration the desired functionality, after preparing the filter paper samples at an original pH of 4, 7 and 11, their color changes when in contact with 3% (*v*/*v*) acetic acid and 35% ammonia were tested. To evaluate their sensitivity to liquids, a 10 µL aliquot of each chemical solution was placed on the samples and the possible color change was observed.

### 2.4. Preparation of pH-Sensitive Formulations and Application on Cotton and Polyester Knits

The conditions for polymeric formulations were similar to those described above for aqueous solutions (Section 2.2). The aqueous base has been replaced by the polymer base, Edolan SN. The above-mentioned indicator combinations and glycerol 16% (*v*/*v*) were dissolved in Edolan SN and placed under vigorous magnetic agitation for 1 h. The solution pH was adjusted to approximately 7 using 5% citric acid or 1 M sodium hydroxide, added dropwise. Before applying to textiles, 1% (*w*/*v*) of Thickener A02 was added to the polymeric solution and agitated by mechanical stirring for 10 min. The pH-sensitive formulations were then stamped on cotton (CO) and polyester (PES), using a stamping table Zimmer Magnet System Plus. Two layers of the polymeric solution were applied to each sample, and all samples were dried at 100 °C for 5 min. A schematic representation of this process is shown in Figure 1.

### 2.5. Microscopic Characterization of pH-Sensitive Textiles

Microscopic characterization was performed according to previous research [29,30]. The main goal was to analyze the deposition of the coating on cotton and polyester fibers. An optical microscope Leica DM750 M (brightfield) (Leica, Wetzlar, Germany) with a high-definition camera was used. For all samples, at least two different zones were observed at a total magnification of 50.

### 2.6. Air-Permeability Assessment

The airflow capable of passing through a given area of the textile samples was analyzed, making it possible to indirectly evaluate the material porosity and its breathability. This parameter was measured with an adaptation of the ISO 9237 standard using an air permeability tester III FX 3300 of TEXTEST Instruments, with a pressure of 200 Pa and a head area of 20 cm^2^. To compare the samples functionalized, control samples of CO and PES were also evaluated. 

### 2.7. Contact Angle Assessment

The analysis of the contact angle was done with a water contact angle test, in which the contact angle between a water drop and the sample surface was analyzed. A volume of 5 µL per water drop and a flow rate of 10 µL/s was used. To carry out this test, a goniometer contact angle system OCA 15 was connected to a digital microscope equipped with a camera. The measurements were done in different portions of the same textile sample. Control knits of CO and PES, before functionalization, were also evaluated.

### 2.8. Evaluation of Samples Sensitivity to Chemicals—Liquid and Gas Detection

To test the ability of the CO and PES samples to detect acidic and alkaline liquids, a 10 µL aliquot of each of the following aqueous solutions was placed on the textiles: 30% (*w*/*v*) citric acid; 30% (*v*/*v*) acetic acid; 10% (*v*/*v*) formic acid; 1 M sodium hydroxide; and 35% ammonia. On the other hand, a set-up was created to assess the sensitivity of the prepared textiles to the presence of gases (alkaline and acid). The set-up constructed consisted of a flask containing 4 mL of the chemical and a 500 mL glass facing downwards (Figure 2). Each sample was left for 5 min, to guarantee the creation of a saturated environment. The solutions used for the gas detection assay were pure hydrochloride acid and 35% ammonia. Image acquisition was done in environmental conditions with a digital camera Canon PowerShot SX530 HS, with 16 megapixels resolution and 4.3 mm focal length.

### 2.9. Statistical Analysis

The statistical analysis of the results obtained throughout this paper was done using the software GraphPad Prism 6.01. The results are presented as the average of the replicates performed, and the respective standard deviation. The results were analyzed using a one-way ANOVA test and multiple comparisons were performed using Šídák’s test. A critical value for significance of *p* < 0.05 was used throughout the study.

## 3. Results and Discussion

### 3.1. Evaluation of Paper Substrates’ Sensitivity to Liquid and Gas Chemicals

Having in mind the end use of these pH-sensitive compounds, after the preparation of the aqueous solutions and application on filter paper, an assay was conducted to confirm the function of the dual sensors and their ability to produce a visible response when an external stimulus is given.

As shown in Table 1, for liquid sensitivity, it is possible to verify that, when placed in contact with solutions of the two extremes of the pH scale, the three sensors can detect them and exhibit a change in color perceptible to the naked eye. It should also be noted that the colors obtained after the addition of acetic acid and ammonia are quite different from the original ones, facilitating the operator’s interpretation. The capability of detecting the two extremes of the pH scale is due to the combination of two pH indicators, with one being sensitive to a lower/more acidic pH and the other to a higher/more alkaline pH. Regarding the use of different pH (4, 7 and 11) for the initial solutions, the response time and the color intensity observed during the assay were similar between samples. The color difference was noticeable within a few seconds, which indicates that the change of this parameter does not influence the sensitivity of the pH indicators, either in contact with acidic or alkaline solutions.

### 3.2. Characterization of pH-Sensitive Textiles

After proving the functional capability of pH double sensors, the compounds were dissolved in a polymeric solution and applied to fibrous substrates by stamping. To analyze the uniform deposition, cotton and polyester-coated fibers were observed using an optical microscope. As shown in Figure 3, before functionalization, the CO fibers were white; however, after the samples were stamped with the chemical sensors their properties shifted and fiber coating can be seen. It is possible to detect a uniform coating of the chemical sensors, even if some areas of the fabric are not completely covered and some naked fibers are visible. This occurred particularly when a solution of methyl orange and bromocresol purple (MO:BP) was used. When it came to PES fibers (Figure 4), compared to the control presented, it was observed that the polymeric solutions were able to provide a homogeneous coating on the knits for all three sensors. In this case, few or no uncoated fibers are observed.

### 3.3. Air Permeability and Contact Angle Evaluation of Cotton and Polyester after Coating

When it comes to CPC there are several parameters to consider, both from the safety and protection point of view and for the comfort of the wearer. Two parameters that must be evaluated are air permeability and the contact angle.

Regarding air permeability, as seen in Figure 5, the addition of the polymeric solution significantly altered the capacity of the air to pass through the knits, particularly in PES. For polyester, there is a decrease of approximately 63% in functionalized samples when compared to the control. MO:BP was the sensor that allowed a higher reduction, both for CO and PES. Given that a fabric’s air permeability is directly proportional to its air porosity these results were as expected, i.e., more porosity equals more permeable fabric [31]. After being coated, the fabrics’ pores are filled, leading to their reduction and impact on air permeability.

As previously mentioned, it is important to have in mind that air permeability cannot be too high to the point of compromising the protective barrier, but it also cannot be too low to ensure comfort. If the permeability is too low, there would be an inhibition of the evaporation resulting from sweat passing through the clothing, leading to accumulation on the inner surface and subsequently discomfort [32]. For better comparison with the literature, the values of air permeability for the samples functionalized with MO:BP were recalculated to different units. As shown in Table 2, the samples obtained a value of approximately 2400–2500 L/min/cm^2^/bar, which is higher than 1115 L/min/cm^2^/bar, the air permeability measured by Havenith et al. (2011) for an NBC suit normally used as protective equipment against chemical threats [33]. However, since the main goal of the developed textiles is to use them as a sensing mechanism on top of another piece of CPC and not as a single barrier, the results obtained for MO:BP samples show a good compromise between the permeability needed for protection and comfort, which can be further reduced as needed by combining it with an equipment with lower permeability.

Furthermore, protective textiles must at least meet the hydrophobicity threshold to prevent the passage of chemicals. Contact angle measurements reflect the ability of a solid sample to repel a liquid, and therefore, higher contact angles are related to a superior repellent effect [34]. A textile surface is hydrophobic if the water contact angle is higher than 90°, and its hydrophobicity increases in proportion with its contact angle [35], until reaching superhydrophobic if it is more than 150° [36]. The contact angle for CO and PES without coating is not represented in Figure 6 since the water drop was immediately absorbed by these samples, making it impossible to measure. Therefore, we can assume that before the application of the coating the textiles exhibited a hydrophilic behavior. The results obtained (Figure 6) showed that, after the functionalization with the halochromic compounds, all the samples can be considered hydrophobic. As mentioned in other research articles, this parameter is of extreme importance in protective equipment [37,38]. In a study performed by Antunes et al. the wettability/hydrophobicity of different fabrics commonly used in PPEs (cotton, jersey (cotton + elastane) and TNT) was evaluated after the application of a coating [39]. The contact angles achieved were similar to the MO:BP_PES and MR:MR_CO samples developed in the present work, with a value around 120°. Similar results in contact angle were presented by Irzmańska et al. in a cured rubber sample for subsequent application to personal protective gloves [40].

Statistical analysis was performed to evaluate existing differences between the same sensors applied on different fibrous substrates and the response of cotton and polyester individually to the functionalization with different sensors. When fixing the sensor and comparing between cotton and polyester samples, the MR:MR double sensor was the only one that demonstrated significantly different behavior when the fibrous substrate was changed, with the others exhibiting similar contact angles. On the other hand, comparing the application of the three sensors on the same textile, only polyester displayed significant behavior alterations with MO:BP_PES having a higher contact angle than MR:MR_PES and MR:BB_PES.

### 3.4. Sensors Detection of Liquid and Gas Chemicals after Coating on Cotton and Polyester

After characterizing cotton and polyester samples stamped with pH indicators, functional validation was performed. To understand if the liquid sensitivity is maintained, a test using three different acids and two bases was carried out. When it comes to CO (Figure 7), a notorious difference has been seen after the textile contacted with alkaline and acidic solutions. With acids, the response is more perceptible for MR:MR and MO:BP. Initially MR:MR is orange and shifts its color to a light red, and MO:BP goes from yellow to orange. Even if we can still see some color change when MR:BB is used, the reaction is less noticeable since the sample changes from dark orange to light red. On the other hand, the same is not true for alkaline solutions. In that case, for all three sensors, the color changes are easily seen by the naked eye and detected by the operator. For both sodium hydroxide and ammonia, MR:MR changed from orange to dark blue/green, MR:BB from orange to yellow and MO:BP from yellow to dark purple. Similar results were obtained for PES knits (Figure 8) and a visible response occurred when the textiles contacted with acids and bases. Additionally, in polyester functionalized with MR:BB, the color change with acids is more perceptible than it was in the CO knits. Thus, compared to the initial results, functional activity was maintained even after changing from a paper substrate to a fibrous one.

The observed color variations were as expected and triggered by a shift in the UV-VIS spectrum of the compounds, i.e., according to the conditions of exposure being acidic or alkaline, pH-sensitive compounds exhibit a different maximum wavelength absorption and consequently a different color [41,42,43,44]. In acid environments, MR strongly absorbs at a wavelength of approximately 515 nm and, due to deprotonation in basic pH this compound has a peak at approximately 431 nm [41]. In the case of MR:BB, the color changes are due to MR’s absorption profile, and to BB, which in acidic solutions absorbs at 433 nm and at alkaline pH shifts to a maximum absorption at 615.5 nm [42]. For the sensor MO:BP, due to a hypsochromic shift, MO changes its maximum absorption wavelength from 508 nm to 466 nm, in acid and basic conditions, respectively [43], and BP strongly absorbs at a wavelength of approximately 430 nm in acidic conditions and shifts to 590 nm in basic solutions [44].

It is important to take into consideration that the colors and sensitivity displayed by the sensors are directly dependent on the textile characteristics, such as fiber composition, structure and shade. As a result, depending on those variables, pH indicators may show slightly different behaviors. Herein, particularly for acidic solutions, the fibrous substrate used had an impact on color change perception, which was more noticeable on PES than on CO. This observation was also made by Van der Schueren and De Clerck when studying the use of pH indicators on cotton and nylon, with the two textiles having different detection responses using the same halochromic compound [19].

The response time of all the sensors in both textiles is immediate, which means the user can act accordingly and take the steps needed to guarantee his safety right after contact with the chemicals. Additionally, comparing the two fibrous substrates and the three double sensors tested, PES functionalized with MO:BP displayed more obvious color changes for both acidic and alkaline liquid chemicals, making it more suitable for application on CPC. Additionally, halochromic pigments have already been applied to a wide variety of materials [25,45,46], which indicates that the functionality is only related to the molecules and is not provided by the substrates used, making this formulation suitable to be adjusted to different types of PPE.

Many chemical threats are often invisible. For example, neurotoxic agents, hemotoxic agents and suffocate chemicals, such as phosgene (CG), poisonous organophosphate and cyanogen chloride are often dispersed in the form of a gas [47,48]. Thus, in a scenario where there is a risk of exposure to hazardous chemicals, the detection of toxic gaseous atmospheres is also of great importance. In that sense, the study developed also aimed to produce a textile capable of sensing chemicals in their gaseous state. For this assay pure hydrochloric acid and 35% ammonia were tested, using the set-up described in Section 2.8. For both CO (Figure 9) and PES knits (Figure 10), there is a significant color change after exposure to acid and alkaline environments. Since the color modification of the textiles for hydrochloric acid and ammonia is notoriously different, it is possible to identify the type of chemical an individual has been in contact with and act accordingly. Furthermore, all three sensors evaluated can produce an optical response to the given stimulus in less than one minute, making them suitable for situations where the wearer needs to have quick action in case of exposure. Comparing the color spectrum, as previously observed for liquids, the variation exhibited by MO:BP is more preeminent, displaying opposite/distant colors before and after exposure to chemicals. Hence, these compounds may be of more interest for future applications since they allow an easier perception of the risk.

When comparing the two textiles used in this study, contrary to what was observed for the sensitivity to liquid substances, no influence of the fibrous substrate was observed for the detection of gaseous environments, with CO and PES responding to external stimuli quite perceptibly. Moreover, gaseous atmospheres appear to produce a more distinguishable response than liquid chemicals, which may be due to the volatile nature of the compounds and the saturation of the environment to which the samples are exposed. For the assessment of liquid sensitivity, textiles are only in contact with a small chemical aliquot, which provides a more localized response that is not extendable to the entire sample.

Along with detection capacity, another advantage of the pH sensors used throughout this study, particularly MO:BP, is their reversibility. As explained above, the halochromic behavior of the textiles is due to a change in the UV-VIS spectrum of pH indicator compounds and, when in contact with a solution of its original pH, for example, a neutral buffer, a reaction should occur and the compound once again shifts its maximum absorption wavelength and returns to its original color [44,49]. A study performed by Hong et al. showed the reversibility of a halochromic fiber embedded with three dyes, namely bromothymol blue, thymol blue and methyl yellow [50]. After initial exposure to acidic and basic agents, the fibers were subjected to a buffer solution (pH 8) and returned to their initial state, demonstrating detection capacity for both acid and alkaline chemical agents even upon continuous cycles of stimulus.

## 4. Conclusions

This study focused on the development of cotton (CO) and polyester (PES) capable of detecting chemical agents potentially dangerous to the user, displaying a visible and easy-to-interpret signal. To achieve materials with this property, different types of pH indicators were used, namely, methyl red (MR), methyl red sodium salt (MR), bromothymol blue (BB), methyl orange (MO) and bromocresol purple (BP). Initially, these indicators were tested in an aqueous state and applied on a cellulose substrate-filter paper. It was found that the aqueous solutions tested were sensitive to bases and acids in the liquid state and a pH of 7 was selected for the preparation of the polymeric formulation. After stamping the double sensors on CO and PES knits, the mechanical and structural properties of the functionalized knits were also analyzed. In line with the results observed for the paper filter, both cotton and polyester were able to detect acids and bases in both liquid and gaseous states. The coating of polyester with MO:BP polymer formulation showed the greatest potential, due to its sensing capability as well as lower air permeability (2412.5 L/min/cm^2^/bar) and hydrophobic behavior (contact angle of 123°). The color shift observed was as expected and triggered by a change in the UV-VIS spectrum of the compounds, consequently altering the substrate’s color. Herein, an optimization of the coating process was seen, enabling the industrial application of the sensors tested for personal protection as an alternative to other time and resource-consuming techniques, such as exhaustion. Moreover, as previously stated, the textile sensor developed is intended to be used on CPC as patches in strategically selected locations such as on the sleeve, near the cuff and on the chest, since those are the most likely points of contact with harmful agents and are easier to be seen by the user.

For future research, it would be valuable to evaluate the durability of the sensors studied on CO and PES, by analyzing wash resistance, abrasion resistance and the resistance of materials to liquid penetration. Furthermore, for application on military CPC and on-site detection, it would be important to study these sensors’ response to substances that mimic the action of the most well-known and used chemical warfare substances, as well as analyze the minimum concentration necessary to trigger a reaction by the material.

## Figures and Tables

**Figure 1 materials-16-02938-f001:**
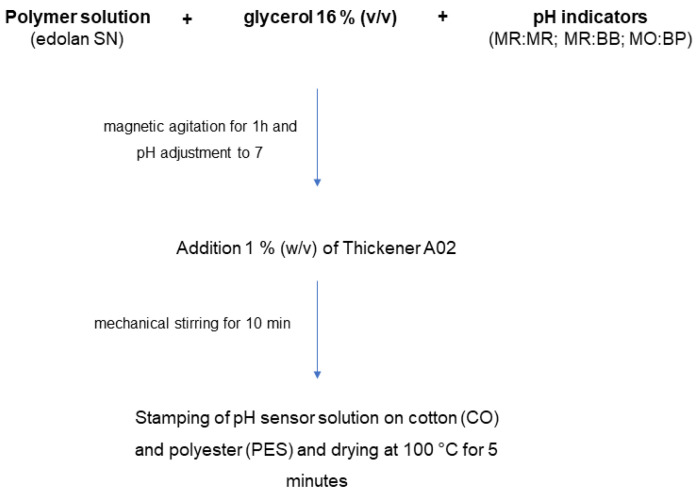
Schematic representation for the preparation of pH-sensitive formulations and application on cotton and polyester knits.

**Figure 2 materials-16-02938-f002:**
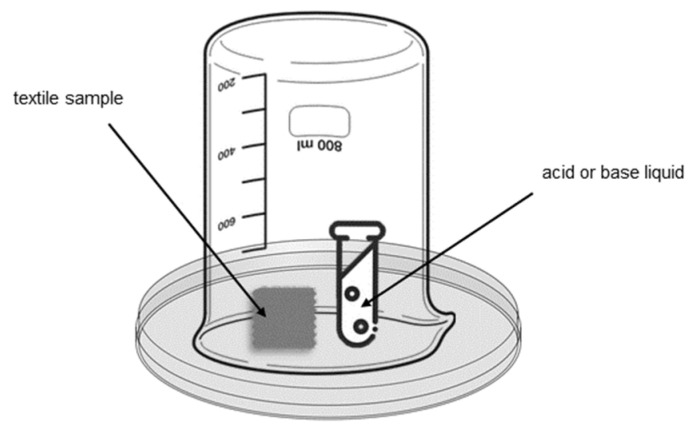
Set-up used in the sensitivity assay of textile samples to gases.

**Figure 3 materials-16-02938-f003:**
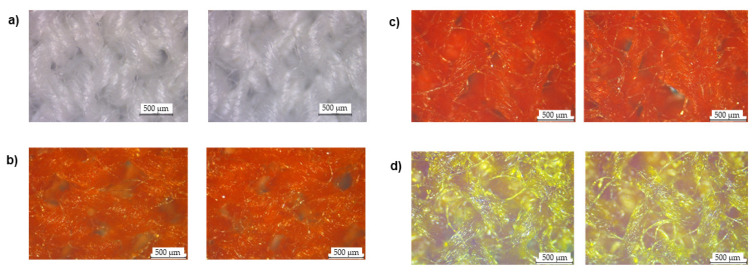
Cotton fibers were observed under a microscope at a total magnification of 50, (**a**) before and after coating with (**b**) methyl red and bromothymol blue (MR:BB), (**c**) methyl red and methyl red sodium salt (MR:MR) and (**d**) methyl orange and bromocresol purple (MO:BP).

**Figure 4 materials-16-02938-f004:**
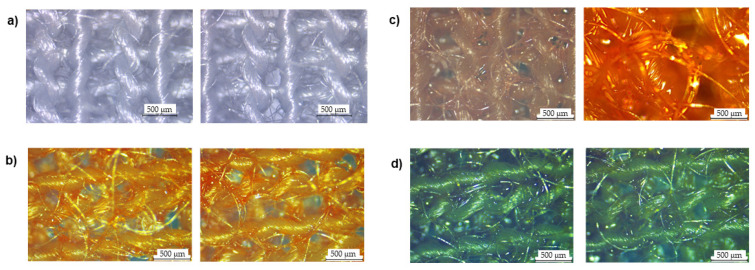
Polyester fibers were observed under a microscope at a total magnification of 50, (**a**) after coating with (**b**) methyl red and bromothymol blue (MR:BB), (**c**) methyl red and methyl red sodium salt (MR:MR) and (**d**) methyl orange and bromocresol purple (MO:BP).

**Figure 5 materials-16-02938-f005:**
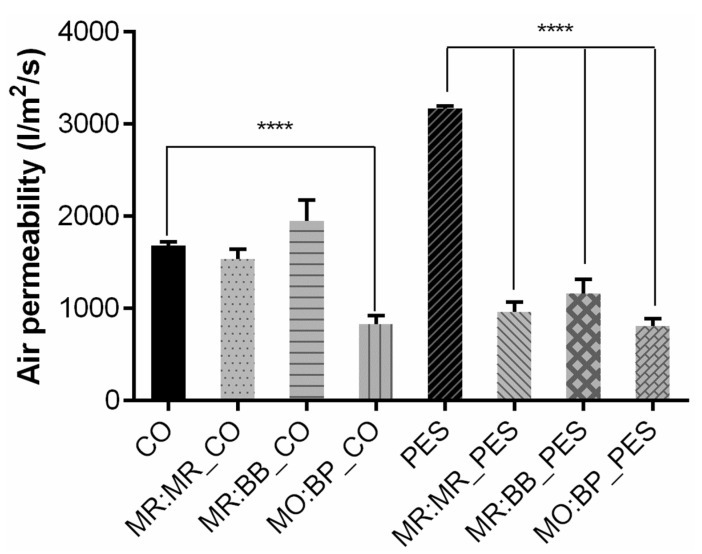
Air permeability measurement of cotton (CO) and polyester (PES) before and after coating with different pH-sensitive formulations: methyl red and bromothymol blue (MR:BB); methyl red and methyl red sodium salt (MR:MR); and methyl orange and bromocresol purple (MO:BP), (n = 5, ±SD), **** *p* < 0.0001 (one-way ANOVA, Šídák’s test).

**Figure 6 materials-16-02938-f006:**
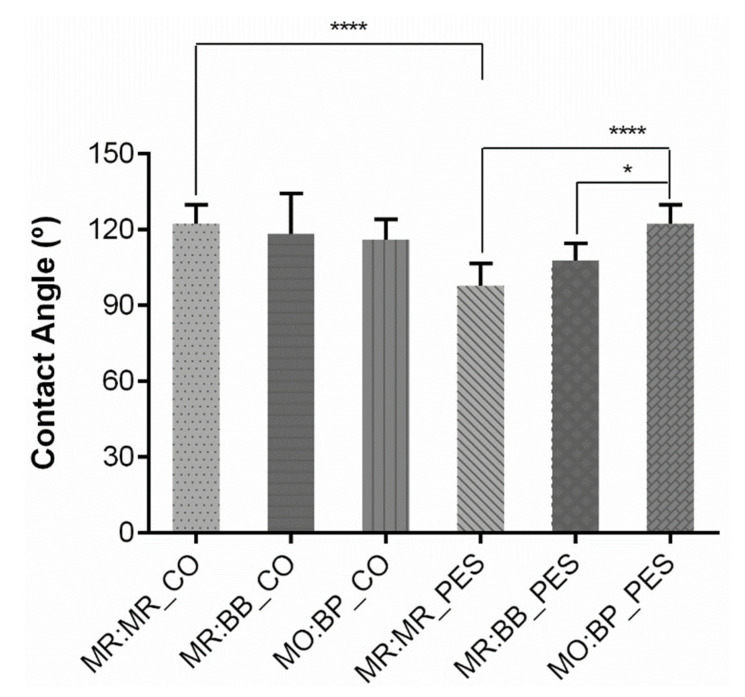
Contact angle measurement of cotton (CO) and polyester (PES) after coating with different pH-sensitive formulations: methyl red and bromothymol blue (MR:BB); methyl red and methyl red sodium salt (MR:MR); and methyl orange and bromocresol purple (MO:BP), (n = 10, ±SD), * *p* < 0.05, **** *p* < 0.0001 (one-way ANOVA, Šídák’s test).

**Figure 7 materials-16-02938-f007:**
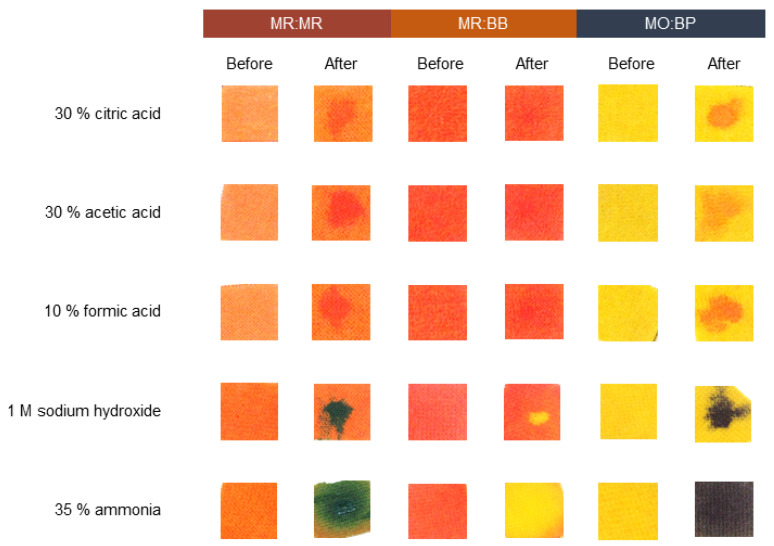
Sensitivity of cotton knits functionalized with different pH sensors, to acidic and basic solutions. The sensors used were: methyl red and bromothymol blue (MR:BB); methyl red and methyl red sodium salt (MR:MR); and methyl orange and bromocresol purple (MO:BP).

**Figure 8 materials-16-02938-f008:**
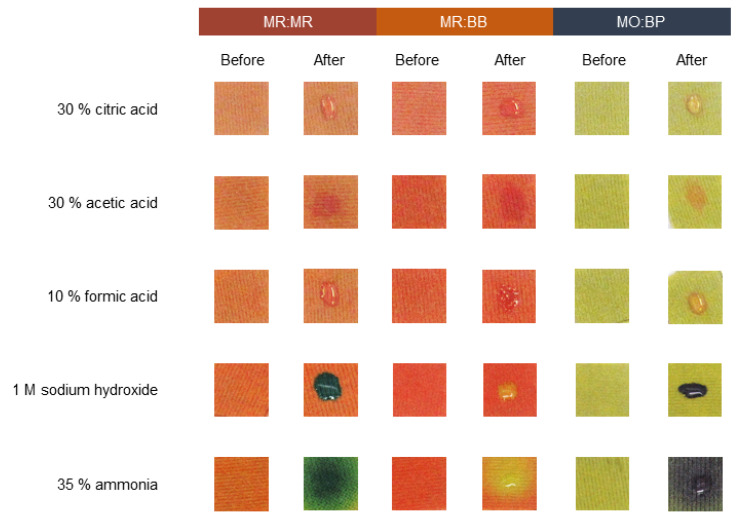
Sensitivity of polyester knits functionalized with different pH sensors, to acidic and basic solutions. The sensors used were: methyl red and bromothymol blue (MR:BB); methyl red and methyl red sodium salt (MR:MR); and methyl orange and bromocresol purple (MO:BP).

**Figure 9 materials-16-02938-f009:**
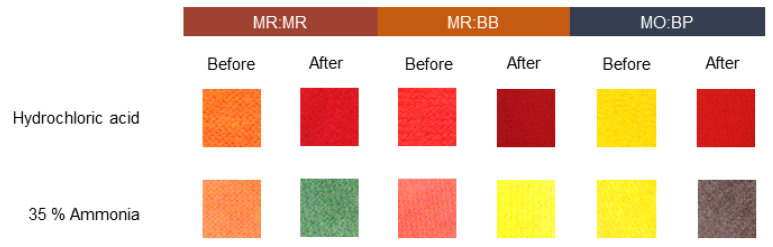
Sensitivity of cotton knits functionalized with different pH sensors to a basic environment created by 35% ammonia and hydrochloride acid solutions. The sensors used were: methyl red and bromothymol blue (MR:BB); methyl red and methyl red sodium salt (MR:MR); and methyl orange and bromocresol purple (MO:BP).

**Figure 10 materials-16-02938-f010:**
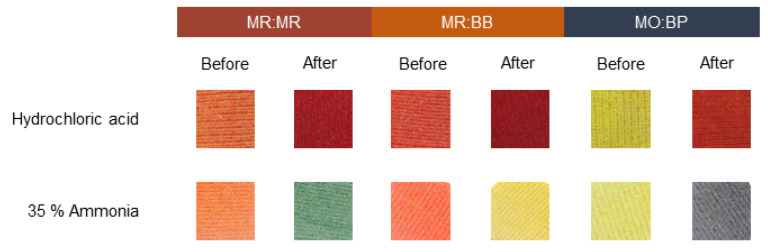
Sensitivity of polyester knits functionalized with different pH sensors to a basic environment created by 35% ammonia and hydrochloride acid solutions. The sensors used were: methyl red and bromothymol blue (MR:BB); methyl red and methyl red sodium salt (MR:MR); and methyl orange and bromocresol purple (MO:BP).

**Table 1 materials-16-02938-t001:** Sensitivity of filter paper dyed with pH-sensitive aqueous solutions to 3% (*v*/*v*) acetic acid and 35% ammonia. The pH-sensitive solutions used were prepared at different pH 4, 7 and 11, and consisted of different combinations: methyl red and methyl red sodium salt (MR:MR); methyl red and bromothymol blue (MR:BB); methyl orange and bromocresol purple (MO:BP).

		3% Acetic Acid	35% Ammonia
MR:MR	pH ≈ 4	✓	✓
	pH ≈ 7	✓	✓
	pH ≈ 11	✓	✓
MR:BB	pH ≈ 4	✓	✓
	pH ≈ 7	✓	✓
	pH ≈ 11	✓	✓
MO:BP	pH ≈ 4	✓	✓
	pH ≈ 7	✓	✓
	pH ≈ 11	✓	✓

“✓”—naked eye color change observed.

**Table 2 materials-16-02938-t002:** Air permeability values of the cotton (CO) and polyester (PES) samples functionalized with methyl orange and bromocresol purple (MO:BP), converted into different units.

Air Permeability Units	MO:BP_CO	MO:BP_PES
L/m^2^/s (@200 pa)	827.4	804.4
L/min (@200 pa)	99.3	96.5
L/min/cm^2^/bar	2482.5	2412.5

## Data Availability

Not applicable.

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
