# Peer review of "Halochromic Textiles for Real-Time Sensing of Hazardous Chemicals and Personal Protection"

_materials, 2023, doi:10.3390/ma16082938_

Round 1

Reviewer 1 Report

Dear author,

The topic presented in the paper is actual and relevant to the research field. 

In order to provide an excellent article to the readers, some parts of it should be improved and I will be listed in the order they appear. 

Pag. 3, 2.1. Materials

- this paragraph does not sound very good, even if the chemicals were purchased from different companies. Please rephrase it in a less commercial way.

- Pag. 4. In paragraph 2.2, it is not clear the connection between the substrates and sensors. Maybe a clear definition of sensors would help.

- Are they any standards for the measurements described at 2.2-2.9 points? If yes, they should be mentioned in the text. 

- Pag. 7.  The text after Table 1...''The symbol “✓” ..'' should be moved somewhere else. 

Conclusions could be more comprehensive regarding the integration of the proposed sensors into the CPC. 

Reviewer 2 Report

The topic of the presented manuscript and also its application are interesting, but it needs strong revisions on both structural and grammatical aspects. Some comments are suggested bellow:

- There are several grammatical errors. It is strongly suggested to double check the text carefully. Editing the manuscript by an at least professional speaker of English is strongly encouraged.

- What is the novelty of work? The novelty and innovation in this study should be mentioned clearly compared with previous studies.

- At section 2.1 Materials, it is required to provide more details about the fabrics’s characteristics.

- In figure 3, it is suggested to add the same image from the raw substrate, as well as in Figure 4.

- To better understanding and comparison, in Figure 6, it is suggested to add the data for the raw (un-coated) cotton and polyester fabrics.

- Enough discussion on the result and discussion section couldn't be seen. It is strongly suggested to improve this part and use the new-published relevant researches to confirm the results.

Reviewer 3 Report

Reviewers' comments:

Manuscript number: materials-2207053

Title: Halochromic textiles for real-time sensing of hazardous chemicals and personal protection.

The manuscript reported on Halochromic textiles for real-time sensing of hazardous chemicals and personal protection. The manuscript needs a detailed editing. It cannot be recommended for publication in the present form. I hope the following points would be helpful for the authors.

The authors need to consider the following comments

- In the Abstract, the authors need to improve with more specific short results and conclusions, i.e. academic novelty or technical advantages.

- In the introduction, the authors do not show the significance and novelty of the work.

- 2.5 Microscopic characterization of pH-sensitive textiles - should be provide more details.

- 2.7 Contact angle assessment – should be improve.

- 3.2 Characterization of pH-sensitive textiles – should be improve.

- Figures 3 and 4 - not clear make clear.

- Main findings should also be provided in conclusions.

- References: make all references in same format for volume number, page number and journal name, because it is difficult to searching and reading.

- Some sentences need reconstruction and the level of English should be improved.

Based on these, I advise the authors to rectify the above mentioned errors and we hope to re-evaluate the revised manuscript.

Round 2

Reviewer 3 Report

The authors revised the manuscript according to the reviewers' comments.